# Development of an Argon Light Source as a Calibration and Quality Control Device for Liquid Argon Light Detectors

**Mehmet Tosun** [1,2,3,*], **Burak Bilki** [1,2,4], **Fatma Boran** [1,5], **Furkan Dolek** [1,5] and **Kutlu Kagan Sahbaz** [1,2,3]

1. Department of Mathematics, Beykent University, Istanbul 34500, Turkey
2. Turkish Accelerator and Radiation Laboratory, Ankara 06830, Turkey
3. Institute of Nuclear Sciences, Ankara University, Ankara 06100 , Turkey
4. Department of Physics and Astronomy, University of Iowa, Iowa City, IA 52242, USA
5. Department of Physics, Cukurova University, Adana 01330, Turkey
* Correspondence: mehmet.tosun@cern.ch

**Abstract:** The majority of future large-scale neutrino and dark matter experiments are based on liquid argon detectors. Since liquid argon is also a very effective scintillator, these experiments also have light detection systems. The liquid argon scintillation wavelength of 127 nm is most commonly shifted to the visible range by special wavelength shifters or read out by the 127 nm sensitive photodetectors that are under development. The effective calibration and quality control of these active media is still a persisting problem. In order to respond to this need, we developed an argon light source which is based on plasma generation and light transfer across a $MgF_2$ window. The light source was designed as a small, portable and easy-to-operate device to enable the acquisition of performance characteristics of several square meters of light detectors. Here, we report on the development of the light source and its performance characteristics.

**Keywords:** liquid argon; plasma light source; scintillation light

## 1. The Argon Light Source

Most future large-scale neutrino and dark matter experiments will rely on liquid argon detectors (see, e.g., [1–5]). For this reason, detectors to measure the scintillation light generated inside liquid argon detectors are needed. The number of photosensors to measure the 127 nm wavelength argon scintillation light is quite limited and usually a wavelength shifter such as tetraphenyl-butadiene (TPB) is employed (see, e.g., [6]). The calibration and quality control of these detectors is still an ongoing problem.

In order to meet this need, we made an argon plasma light source that produces light with a wavelength of 127 nm. The argon light was transferred to the outside of the light source body through a $MgF_2$ window. We made the body of the light source from polyoxymethylene and used titanium wires as the electrodes for the light source. The light source was put under a vacuum of $5 \times 10^{-6}$ mbar and flushed a few times with high purity argon prior to be put in operation. The final filling was done to the target pressure and the chamber was sealed. The operating voltage and pressure were scanned in order to obtain the optimal operating conditions.

Figure 1 shows a picture of the light source in operation. The plasma light can be seen through the $MgF_2$ window. The final filling still contains ppm levels of contaminants which limit the fraction of the 127 nm light. In order to identify the optimal operating conditions, the argon pressure was scanned from 1000 mbar to 2000 mbar in steps of 100 mbar; the operating high voltage was scanned from 2600 V down to the point where the light is lost (usually around 1200 V) in steps of 100 V; and the average spectrum of the light was measured. Figure 2(left) shows a sample average spectrum which shows the argon and impurity peaks in the 200–1000 nm range. The peaks are identified, and the relevant intensity integrals are calculated. Figure 2(right) shows the intensity integrals due to argon

emissions (red squares) and impurity emissions (black circles) for the operating conditions probed. The largest fraction of argon emissions is identified to be at a 1300 mbar pressure and 2600 V high voltage. These conditions were taken as the nominal operating conditions.

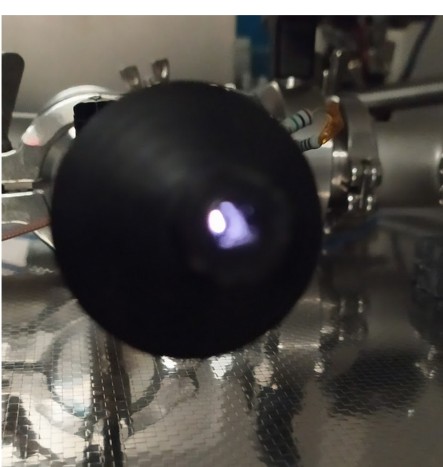

**Figure 1.** A picture of the light source during operations. The front part of the light source is pictured. The light through the $MgF_2$ window is visible.

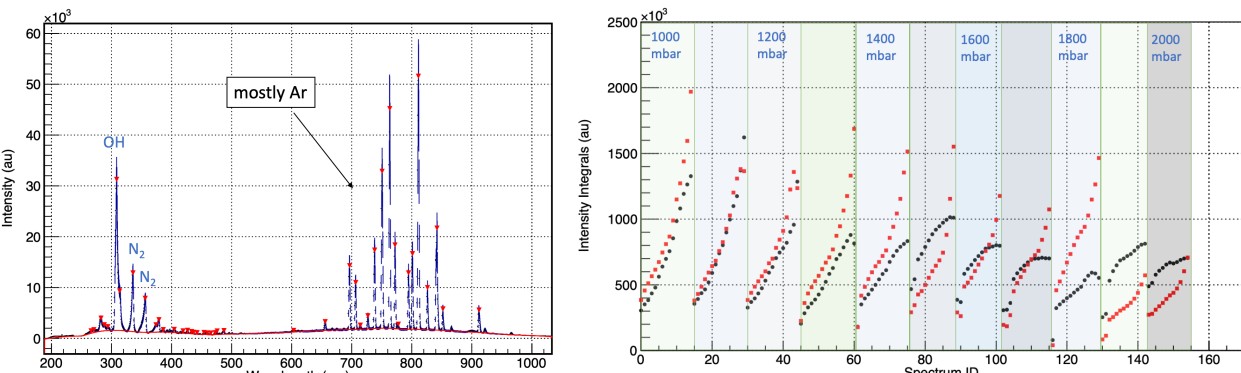

**Figure 2.** A sample average spectrum which shows the argon and impurity peaks in the 200–1000 nm range (**left**) and the intensity integrals due to argon emissions (red squares) and impurity emissions (black circles) during the high-voltage and pressure scan (**right**).

## 2. Validation of the Light Detectors

In order to measure the 127 nm wavelength light, 0.2 mg/cm$^2$ TPB was coated on the 3 mm × 3 mm windows of silicon photomultipliers (SiPMs) (KETEK PM3325-WB-D0). In order to validate the performance of the TPB-coated SiPMs, an assembly of two SiPMs looking at each other and separated by 12 cm was constructed. The assembly was placed in a stainless-steel tube, and a test chamber housing temperature sensors, a pressure transmitter, an LED strip, a cable feedthrough, gas and vacuum connections, and a camera was constructed. The test chamber was used to liquify high-purity argon gas in a liquid nitrogen bath.

The test chamber was put under vacuum down to $8 \times 10^{-6}$ mbar and then filled with high-purity argon gas up to 1300 mbar quickly. The argon gas was then liquified by filling the outer bath with liquid nitrogen. The filling of the liquid nitrogen bath was done manually, continuously monitoring the chamber pressure and adding argon gas. The liquification was also observed through the camera. Figure 3(left) shows a camera image of the liquid at the bottom of the chamber. The entire liquification period was recorded as a video.

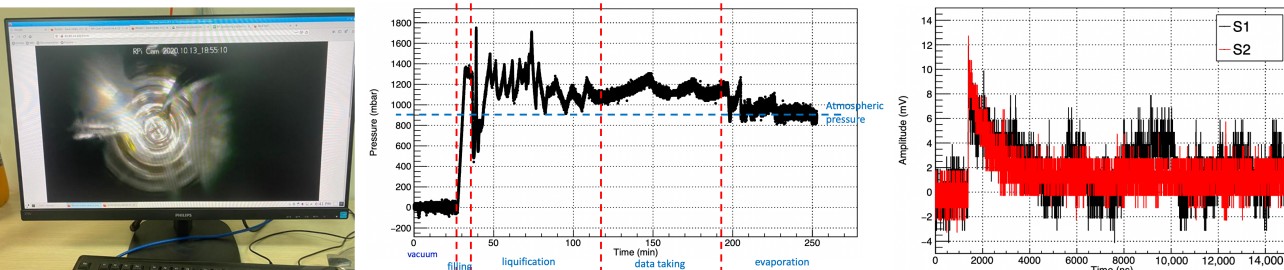

**Figure 3.** A camera image of the liquid argon at the bottom of the chamber (**left**), the chamber pressure as a function of time through the operations (**center**) and the waveforms from a single cosmic muon event recorded by both TPB-coated SiPMs (**right**).

Once the SiPM assembly was completely submerged in liquid argon, the LED was turned off and the measurement with the SiPMs started. The data taking was triggered by the coincidence of SiPM signals above the single photon level. The chamber pressure was continuously monitored, and the liquid nitrogen bath was refilled once the chamber pressure increased up to around 1300 mbar level. The operation was smooth, and the purity of the liquid argon allowed data taking for approximately 60 min. Figure 3(center) shows the pressure as a function of the operation time, starting from the vacuum stage until the evaporation stage. Figure 3(right) shows the waveforms of the two TPB-coated SiPMs, denoted as S1 and S2, for a cosmic muon event.

The average waveform of the cosmic muon signals was calculated and fit to the sum of two exponentials and a constant as shown in Figure 4. The two time constants corresponded to the intermediate and slow components, $\tau_{int}$ and $\tau_{slow}$, of the argon scintillation. The fit results were obtained as $\tau_{int} = 272$ ns and $\tau_{slow} = 1.26$ µs. The time constants were comparable with the values obtained with larger-scale test setups (see, e.g., [7]). It should be noted here that the results were preliminary, with a partial fit, and did not include further calculations such as deconvolution. Therefore, the intermediate component in particular had a very large error margin. The argon purity can be assessed by observing the slow component of scintillation and is sufficiently good for this size of a chamber. As a result of the cryogenic tests, the TPB-coated SiPMs were validated to be used to measure 127 nm argon scintillation light.

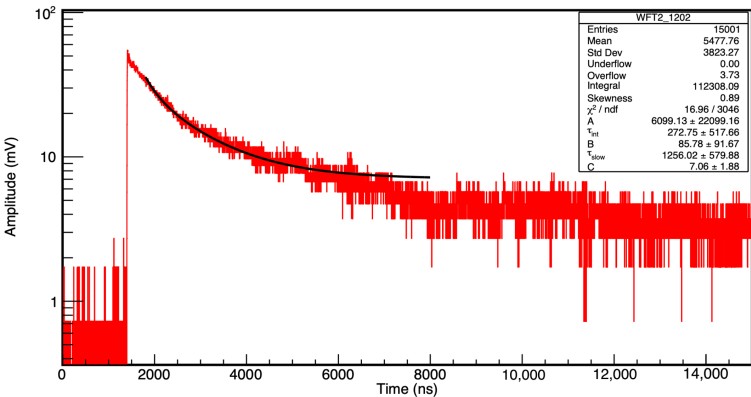

**Figure 4.** The average waveform of the cosmic muon signals and the fit to the sum of two exponentials and a constant.

## 3. Validation of the Light Source

A vacuum-tight test assembly was constructed in order to validate the performance of the argon light source. The exit window of the light source was coupled to a custom flange. Opposite to the light source window was a single SiPM. Another single-SiPM assembly was made with a SiPM with its window coated with TPB. Figure 5 shows the overlaid signals measured with the clean (top) and the TPB-coated (bottom) SiPM looking directly

at the light source under vacuum. The data were recorded with self triggering on the light pulses 20 mV above baseline. The main pulse for the clean SiPM was mostly due to the impurities in the argon, and partly due to the red–infrared emission of argon. Compared to the clean SiPM-overlaid signals, the height of the triggering pulse for the TPB-coated SiPM was decreased and the readout window was populated with many additional pulses suggesting a nearly continuous emission of 127 nm light, which was not visible with the clean SiPM.

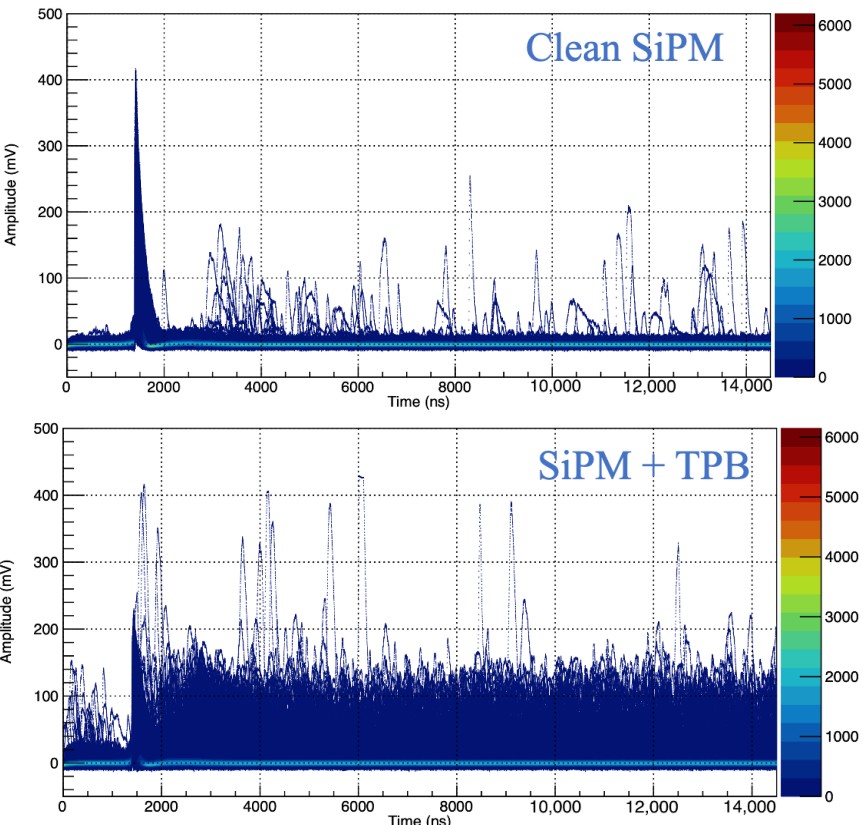

**Figure 5.** The overlaid signals measured with the clean (**top**) and TPB-coated (**bottom**) SiPM looking at the light source under vacuum. The SiPMs were placed right across the $MgF_2$ window of the light source in a vacuum assembly. The data taking was triggered with the SiPM signals themselves slightly above the single avalanche threshold. The trigger was timed to be around 1500 ns and all the waveforms were overlaid to make the plots.

Figure 6(left) shows the number of pulses with peak amplitudes above 30 mV in the 15 μs window per triggered event. The triggered events with the clean SiPM mostly contain single pulses with peaks above 30 mV and the number of two or more peaks is significantly reduced. For the case of TPB-coated SiPM, the number of pulses in the readout window with peaks larger than 30 mV is much higher. As the only difference was the introduction of the TPB on the SiPM window, which simply increased the sensitivity to 127 nm light, the operation of the light source was validated.

Figure 6(right) shows the full width at half-maximum (FWHM) of all the pulses in the 15 μs readout window for the clean and TPB-coated SiPMs. The majority of the pulses have less than 500 ns width. On the other hand, the TPB-coated SiPM pulses have an accumulation around 800 ns. Figure 7(left) shows an example of the pulse with an FWHM less than 500 ns , and Figure 7(right) shows an example with an FWHM larger than 500 ns for the TPB-coated SiPM. The wider pulses are attributed to the 127 nm light. The 127 nm light seems to be originating in bursts within which the individual pulses are a few nanoseconds apart. The time structure of the light is under further investigation.

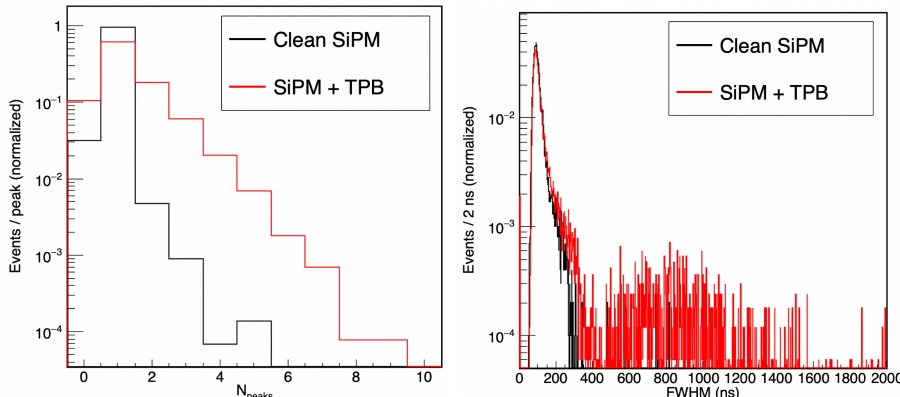

**Figure 6.** The number of pulses with peak amplitudes above 30 mV per triggered event (**left**) and the full width at half-maximum of all the pulses in the 15 µs readout window (**right**) for the clean and TPB-coated SiPMs.

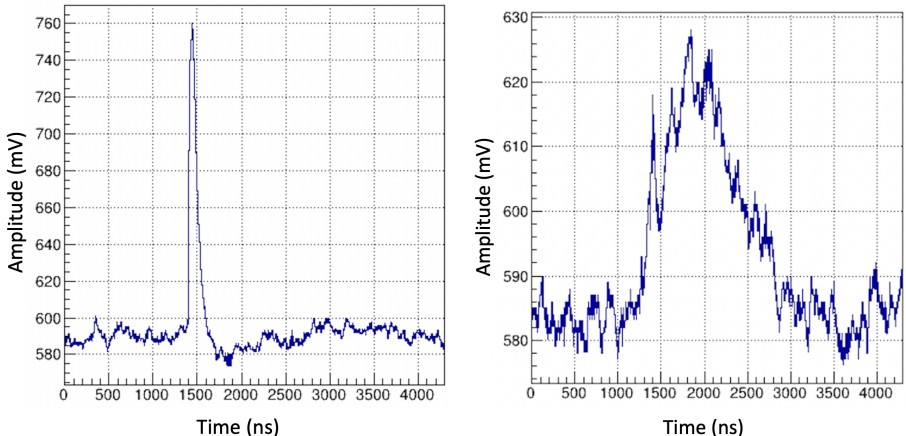

**Figure 7.** Example pulses with full width at half-maximum less (**left**) and larger (**right**) than 500 ns for the TPB-coated SiPM.

## 4. Conclusions

An argon light source envisaged to be a practical calibration and quality control device for liquid argon light detectors was developed. The preliminary characterization of the device indicated that 127 nm argon scintillation light was transferred through the MgF$_2$ window and could be identified with its specific waveform. The complete characterization of the light source is underway. The complete pulse shape discrimination, intensity stability and single filling lifetime are under investigation.

**Author Contributions:** Data curation, M.T.; Investigation, M.T., K.K.S., F.B., F.D.; Project administration, B.B. All authors have read and agreed to the published version of the manuscript.

**Funding:** This work is supported by Tübitak grant no 118C224.

**Data Availability Statement:** The data presented in this study are available on request from the corresponding author.

**Conflicts of Interest:** The authors declare no conflict of interest.

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
