# Peer review of "Development of an Argon Light Source as a Calibration and Quality Control Device for Liquid Argon Light Detectors"

_instruments, doi:10.3390/instruments6040045_

Round 1

Reviewer 1 Report

Development of an Argon Light Source as a Calibration and Quality Control Device for Liquid Argon Light Detectors

The proceedings is well written and clearly introduces the motivations and the results of the ongoing development and characterization of an Argon light source.  It lacks sometimes technical  details which will help the reader to understand the details. I guess this is due to the page limitations?  Part of the work is also still ongoing and it will be interesting to follow up in the future. 

My main comments are below. 

Section 1. The Argon Light Source

No information on the SiPM used is provided. Please add. 

I agree on the procedure for [2. Validation of the Light Detectors], but I think it needs more work. 

The fit of Figure 4 is very poor, and I dont understand why the fit range is limited to [2000, 3000] ns, where, clearly, there is: A) no power to distinguish the two components; B) no capability to resolve a ~1 us tau. The fit results are in fact tau_int = 270±415 ns  and  tau_slow = 1311±897 ns (with huge errors) and it is wrong in my opinion to make ANY claim based on this fit. 

If you want to distinguish the 2 components (which are however clearly visible), you should extend the fit range up to 8000 ns (and add a constant term to account for the flat part). 

More on the intermediate component: there is evidence for a delayed emission from TPB, which should explain it (see Segreto, or ARIS or DarkSide-50). However, it was measured in the past using PMTs (also in the quoted reference), while here you are using SiPMs. SiPMs typically have a slower single PE pulse compared to PMTs, possibly comparable to tau_int. Therefore, the measurement of tau_int may be larger compared to earlier ones if you don't do any deconvolution. Please comment on the SiPM signal shape if this effect is believed to be negligible. 

General: 

Please fix two occurrences of "15 ms" where it should be "15 us". 

Author Response

On behalf of the authors, I would like to thank the referee for the constructive comments and suggestions. Please find my replies below:

Development of an Argon Light Source as a Calibration and Quality Control Device for Liquid Argon Light Detectors

The proceedings is well written and clearly introduces the motivations and the results of the ongoing development and characterization of an Argon light source.  It lacks sometimes technical  details which will help the reader to understand the details. I guess this is due to the page limitations?  Part of the work is also still ongoing and it will be interesting to follow up in the future. 

My main comments are below. 

Section 1. The Argon Light Source

No information on the SiPM used is provided. Please add. 

 ========= Information added.

I agree on the procedure for [2. Validation of the Light Detectors], but I think it needs more work. 

The fit of Figure 4 is very poor, and I dont understand why the fit range is limited to [2000, 3000] ns, where, clearly, there is: A) no power to distinguish the two components; B) no capability to resolve a ~1 us tau. The fit results are in fact tau_int = 270±415 ns  and  tau_slow = 1311±897 ns (with huge errors) and it is wrong in my opinion to make ANY claim based on this fit. 

If you want to distinguish the 2 components (which are however clearly visible), you should extend the fit range up to 8000 ns (and add a constant term to account for the flat part). 

========= All agreed. Fit range was extended up to 8000 ns.

More on the intermediate component: there is evidence for a delayed emission from TPB, which should explain it (see Segreto, or ARIS or DarkSide-50). However, it was measured in the past using PMTs (also in the quoted reference), while here you are using SiPMs. SiPMs typically have a slower single PE pulse compared to PMTs, possibly comparable to tau_int. Therefore, the measurement of tau_int may be larger compared to earlier ones if you don't do any deconvolution. Please comment on the SiPM signal shape if this effect is believed to be negligible. 

========= Yes, we are aware of this. Added a one line comment.

General: 

Please fix two occurrences of "15 ms" where it should be "15 us".

========= Done.

Reviewer 2 Report

Minor textual comments

L15: 127 mn -> please, use non-breakable space between number and units, e.g. LaTeX 127~nm

L30,L86: same as above. Perhaps, there are more cases like that

L97: Fig. 7 -> Figure 7. For consistency with the rest of paper

Figure 2: please, mention right plot in the caption

In general, the figure captions are very short and a bit cryptic.

Detailed comments

Figure 1: Please, expand caption with more explanation of what is on the picture

Figure 3: caption of the right plot. “a cosmic muon event” is it really a single event plot or a distribution measured in cosmic events? Please, rephrase.

Figure 4 is not referenced in the text. Please, provide more details in the caption.

Figure 5: Please, provide a more detailed caption. What is plotted on the z-axis (shown with color)? This plot looks like amplitude vs time to me… Why the color is needed?

References section: I think it is too short, there are must be more studies performed in this area of research. Otherwise, the statement in the “Introduction” section that the calibration and quality control of argon light source is an ongoing scientific problem is too strong…

Author Response

On behalf of the authors, I would like to thank the referee for the constructive comments and suggestions. Please find my replies below:

Minor textual comments

L15: 127 mn -> please, use non-breakable space between number and units, e.g. LaTeX 127~nm

========= Done.

L30,L86: same as above. Perhaps, there are more cases like that

========= All fixed.

L97: Fig. 7 -> Figure 7. For consistency with the rest of paper

========= The convention of writing “Figure” if it starts the sentence “Fig.” otherwise was followed.

Figure 2: please, mention right plot in the caption

 ========= Done.

In general, the figure captions are very short and a bit cryptic.

 ========= Tried to improve a bit.

Detailed comments

Figure 1: Please, expand caption with more explanation of what is on the picture

========= Done.

Figure 3: caption of the right plot. “a cosmic muon event” is it really a single event plot or a distribution measured in cosmic events? Please, rephrase.

========= Done.

Figure 4 is not referenced in the text. Please, provide more details in the caption.

========= Done.

Figure 5: Please, provide a more detailed caption. What is plotted on the z-axis (shown with color)? This plot looks like amplitude vs time to me… Why the color is needed?

 ========= Done. These are the accumulations of the waveforms on top of each other to see the picture for the entire dataset at once. So, they are amplitude vs time with the z axis color codes indicating how much a particular (time, amplitude) coordinate is populated.

References section: I think it is too short, there are must be more studies performed in this area of research. Otherwise, the statement in the “Introduction” section that the calibration and quality control of argon light source is an ongoing scientific problem is too strong…

 ========= Indeed. Tried to expand the list a bit still keeping e.g. saying that these are only examples.